Journal of
open psychology data

DATA PAPER

]u[ ubiquity press

# A New Procedure and Stimulus Set for Examining Cross-Modality Mental Rotation

**JOSHUA E. WOLF** (ID)

**MELISSA LARSEN** (ID)

*Author affiliations can be found in the back matter of this article

## ABSTRACT

We validated the use of 3-D printed Shepard and Metzler-style shapes in a simultaneous cross-modal (vision & touch) mental rotation procedure. Participants compared a visually presented 3-D shape to a 3-D shape they could only feel to determine if the shapes were the same. Participant response time and error rate demonstrated the expected linear increase as the angular disparity of the 3-D printed shapes increased. We expect the freely available data and stimuli from the procedure will be useful to researchers studying both traditional mental rotation and cross-modality mental rotation with complex, highly adaptable, and easy to create shapes.

**CORRESPONDING AUTHOR:**

**Joshua E. Wolf**

Department of Life Science: Psychology and Animal Behavior Programs Carroll University, 100 N East Avenue, Waukesha, WI 53186, USA

jwolf@carrollu.edu

**KEYWORDS:**
Mental Rotation; Tactile; 3-D Printing; cross-modal

**TO CITE THIS ARTICLE:**

# (1) BACKGROUND

Apart from providing a robust demonstration of the temporal processes of human cognition, mental rotation, first reported by Shepard and Metzler (1971), has been the starting point for more than 8,400 published studies attempting to understand the complexities of human cognition. The original research required participants to indicate as quickly as possible whether two perspective line drawings of shapes comprised of 10 cubes arranged with 3 right-angle arms were the same (i.e., same shapes presented in rotated orientations), or different (i.e., mirrored shapes that could not be rotated to match). As the angular disparity increased between the two shapes the participants' response time displayed a linear increase leading to the now widely accepted conclusion that the process of mental rotation of shapes, objects, or even letters takes time, and more importantly, that this mental process of rotation can be measured. These shapes and those inspired by the original research have been used to determine which brain areas are most active during mental rotation tasks (Alivisatos & Petrides, 1997) and to investigate the many factors related to mental rotation performance such as a participant's age (Berg et al., 1982; Iachini et al., 2019), sports training and practice with mental rotation (Ozel et al., 2002; Schmidt et al., 2016; Kail, 1986) or even a participant's sex (Metzler & Shepard, 1974; Tapley & Bryden, 1977; Robert & Chevrier, 2003).

While this is not an exhaustive review of the factors which influence mental rotation performance, many of the previously mentioned tasks utilize procedures where participants receive all information about the shapes visually. However, given that our sensory experience with the environment is usually multi-modal (i.e., combinations of sensations such as touch, sight, & smell) many researchers are interested in how touch, or touch in addition to vision, contribute information needed to complete mental rotation. Tactile mental rotation, used here to refer to any mental rotation task where participants touch either the sample or target shape, has been used to investigate how blind and sighted participants perform mental rotation (Marmor & Zaback, 1976; Carpenter & Eisenberg, 1978), how tactile mental rotation compares to visual rotation (Robert & Chevrier, 2003; Norman et al., 2004; Lawson, 2009; Gardony et al., 2014; Tivadar et al., 2019), and how practice effects in one modality transfer to the other (Toussaint et al., 2012). Overall, tactile mental rotation appears to be governed by the same set of rules that apply to visual mental rotation first observed by Shepard and Metzler (1971) such as a linear increase in response time and error rate as the angular disparity increases between the sample and target object. However, despite the interest in and comparison between the two distinct types of mental rotation there is much less research that looks at cross-modality mental rotation, specifically mental rotation

where participants interact with one object visually and the other through touch. Volcic et al. (2010) asked participants to touch a simple 3-D object consisting of a long and short bar joined by 90° angles and decide if the subsequently presented image of a shape was the same or different. Participants in this study were presumably able to mentally rotate one of the sensory representations of the shapes they could touch only and see only, respectively, because as in other mental rotation research, as the angular disparity between the shapes increased, response times and error rate did as well. While this study supported the idea that cross-modal mental rotation was possible, the shapes used were far simpler than those used in the original Shepard and Metzler (1971) study. Additionally, in the procedure utilized in the Volcic et al., (2010) study, participants were allowed to first touch the tactile stimulus and push a button to display the image of the test stimulus when they were ready. This allowed participants to compare the visual image with an already formed mental image of the tactile shape. The purpose of the current study was two-fold. First, we wanted to validate a new simultaneous tactile mental rotation procedure utilizing 3-D printed versions of shapes inspired by the original line drawings in the Shepard and Metzler (1971) study. Second, we wanted to validate the use of 3-D printed versions of the 3-D images in the repository created and validated by Ganis and Kievit (2015) for use in a simultaneous cross-modal mental rotation procedure.

# (2) METHODS

## 2.1 STUDY DESIGN

The data collected for the current paper comes from in-person, face-to-face sessions between one research assistant and one participant. Research assistants collected response time and participant answers ("Same" or "Different") manually during each session. Each participant received two Practice Trials where feedback was provided followed immediately by 32 Test Trials where different shapes (i.e., not those used during practice) were used and no feedback was given. The study validated the use of eight shape pairs broken into two subsets. Participants therefore saw four shapes during the experiment and saw each shape eight times including four Same and four Different Trials at each of the four angles of rotation (0°, 50°, 100°, & 150°).

## 2.2 TIME OF DATA COLLECTION

All data collection occurred between October 1, 2019 through November 21, 2019.

## 2.3 LOCATION OF DATA COLLECTION

Data collection occurred at Carroll University in Waukesha, Wisconsin, United States of America.

## 2.4 PARTICIPANTS

Thirty-seven college undergraduate students (31 females, 5 males, 1 non-specified; mean age = 19.05 years, SD = 1.14 years) served as participants. Participants chose this study from a list of available undergraduate research participation opportunities and were compensated with course credit. The data from two participants was removed for failure to understand the task (i.e., accuracy below chance levels). All participants signed an informed consent prior to the start of the study. All aspects of the procedure and testing were approved by Carroll University's Institutional Review Board. IRB # 19–025.

## 2.5 STIMULI

Ganis and Kievit (2015) created a set of 48 new 3-D computer images like those used in the original mental rotation research (i.e., cubes connected by their faces and a series of right angles) by Shepard and Metzler (1971) to increase the stimuli availability for computer-based mental rotation research. These shapes varied in both the number of cubes used to create the shape (between 8 and 11 cubes) and the orientations of the arms from which the shapes were created (see Ganis & Kievit, 2015 for the full list of shapes). For example, a shape could be comprised of 8 cubes and have the end arms pointing in the same direction (e.g., up), could be built from 11 cubes and have one of the end arms point up and the other end arm oriented at a 90° angle to the right, or anywhere in between. We used Tinkercad, a free online 3-D printing software, to create a repository of 3-D printing STL files of the 48 Ganis shapes and tested 8 of the shapes in a cross-modal mental rotation study. We intentionally selected two of each type of shape based on the number of cubes (e.g., 2 shapes comprised of 8, 9, 10, or 11 cubes) and shapes with a wide variety of arm positions. All printing was completed using an Ultimaker 2 3-D printer. The cubes used to build each shape measured 18 × 18 × 18

mm. We printed two copies of each shape for use as the sample and the test shape on Same Trials and one copy of the mirrored reverse to be used on the Different Trials. Each shape was numbered (1–8) for identification and had a piece of 3M™ Dual Lock Reclosable Fastener on the underside of the longest arm of the shape to securely hold each shape on the apparatus and to ensure the same angle and position for the shapes across trials. The shapes were separated into two subsets. Each shape in a subset was presented at each of the four angles (0°, 50°, 100°, & 150°) for Same and Different Trials for a total of 32 trials for each participant. Figure 1 shows two of the shapes used in the validation study.

### Apparatus

To present the stimuli, we created a chamber (47 × 34 × 19 cm) from painted plywood that allowed for the visual display of the sample shape and the touch only presentation of the test shape. The apparatus (see Figure 2 & 3) was open on both ends and had a piece of fabric attached to the participant side to prevent the shape being seen. The top of the chamber had a piece of 3M™ Dual Lock Reclosable Fastener where the sample shape was placed during the trials. The sample shape area had a visual angle range of 6.7° × 12.8° – 9.5° × 23.5° when viewed from 12 inches depending on the size (dependent on number of cubes) of the current shape. A response board, made from a white corrugated twin wall plastic sheet, marked with the test shape presentation angles (see Figure 3) fit inside the chamber and was accessible from the experimenter side of the apparatus. The response board also had a piece of 3M™ Dual Lock Reclosable Fastener for attaching the test shapes at the different angles. A Rubik's cube speed cubing timer (SPEED STACKS ® G4 Pro Timer) was attached to the participant side of the response board and was used to record response time on each trial. Contact from both

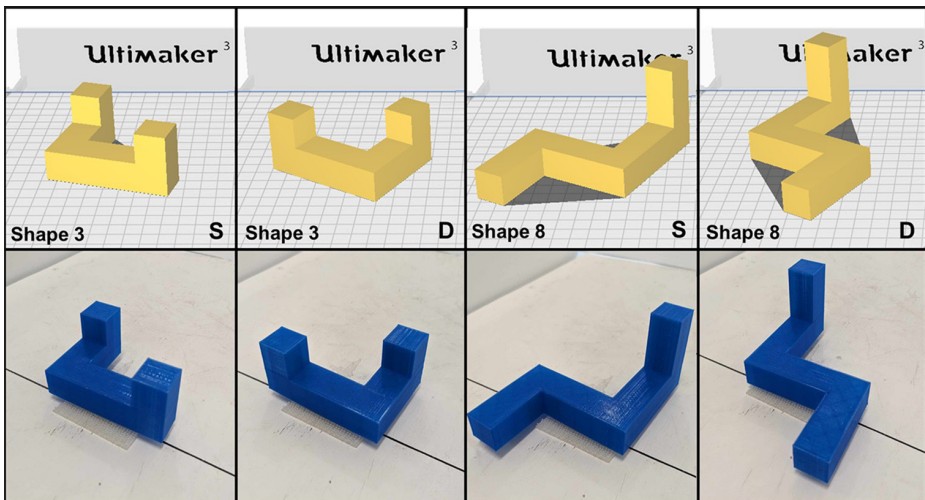

**Figure 1** Figure 1 displays the STL files (top row) for two of the 3-D shapes used in the validation study above the image of the printed shape (bottom row). The shapes used for the sample and Same Trials are denoted with an "S" and the mirror reversed shapes used for Different Trials are denoted with a "D".

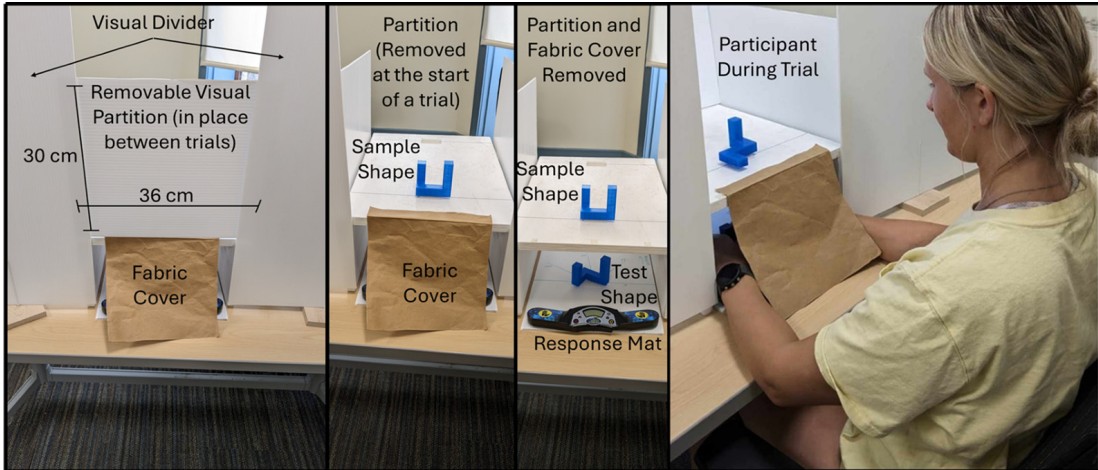

**Figure 2** The three left panels of Figure 2 display the apparatus setup at various stages of a trial with some components labeled and/ or removed for clarity. The right panel displays a participant during a trial.

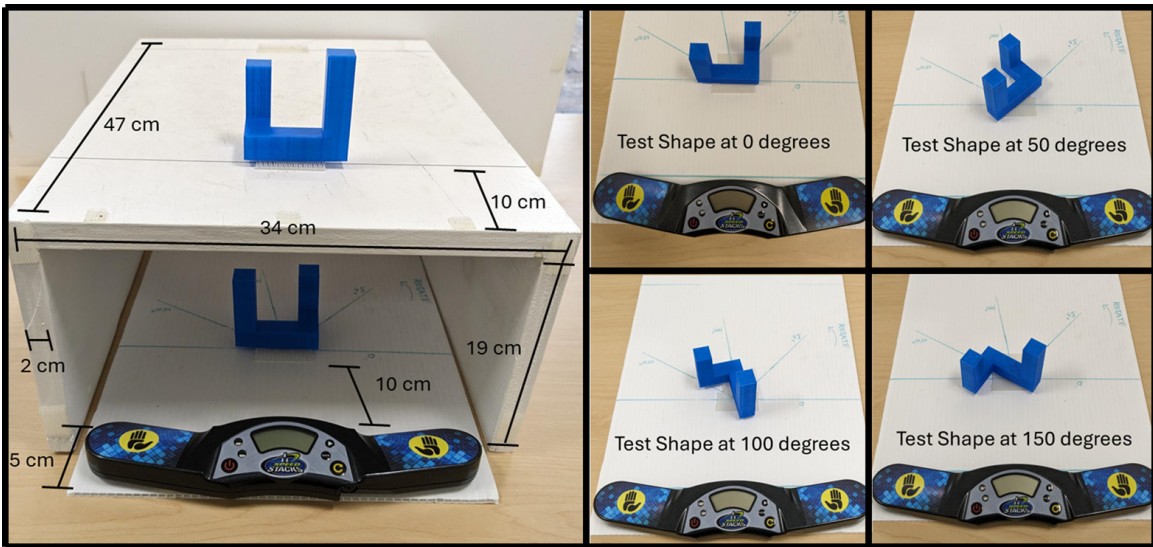

**Figure 3** The left panel of Figure 3 displays the apparatus dimensions and the trial set up for Shape 7 on a Different Trial with the test shape rotated to 0°. The right panel displays Shape 7 at each of the 4 angles of rotation.

hands was required to zero out the stopwatch and when participants removed their hands to touch the shape the stopwatch started and ran until both hands were placed back on the timer. The apparatus also had a large visual divider to prevent participants from seeing anything but the sample shape on top of the chamber. A removable visual partition, made of white corrugated twin wall plastic, prevented participants from seeing the sample shape area during the set up between trials.

## Procedure

After completing their informed consent, participants were shown a sample set of 3-D printed shapes (different from those used in the Test Trials) to familiarize them with the type of shapes used in the experiment as well as to demonstrate the difference between Same and Different Shapes. The shapes included a Sample Shape, a Test Shape that was the same as the Sample Shape, and a Test Shape that was the mirror-opposite of the Sample

Shape. The experimenter demonstrated how the Same Shape was still the same as the Sample Shape even if it had been rotated while the Different Shape would always be a mirrored opposite, even if the angle of rotation was the same.

After seeing the example shapes, participants were seated in a chair and positioned approximately 8 inches from the edge of the table on which the apparatus stood so they could easily reach the test shape inside. Then the experimenter read from the script (freely available @ https://osf.io/rgvf7/) to ensure all participants received identical instructions. Participants were instructed to simultaneously touch the test shape hidden in the response chamber while looking at the sample shape on top of the response chamber and say as quickly as possible if the test shape was the "Same" or "Different". All trials began with the participant inserting both hands into the response chamber and placing their hands on the timer. This hand placement prepared the stopwatch

to begin timing as soon as the participants hands were removed. After the experimenter confirmed that the stopwatch had been reset the partition obscuring the participants view of the sample shape was removed and the participant was able to touch, but not see, the test shape under the apparatus while simultaneously viewing the sample shape on top of the apparatus. They were instructed to return their hands to the timer and say their answer aloud as quickly as possible. The stopwatch ran until both hands were placed back on the timer. The experimenter recorded the participant answer and the response time for each trial manually. Experimenters recorded the first verbal response produced by the participant even if they changced their mind. Each participant first completed two Practice Trials with verbal feedback followed by 32 Test Trials where no feedback was given. The Test Trials were identical to the Practice Trials except that the shapes used were completely novel and no feedback was given. Participants completed 32 trials that ended as soon as there was a response or were cut off at 30 seconds. Each trial, including the 30-s average inter-trial interval and the trial itself took approximately 1 minute to complete.

## 2.6 QUALITY CONTROL
### Data Inclusion
Any responses longer than 30 seconds were removed from the analysis and data from participants 5 and 11 were removed because they gave the incorrect response on greater than half the trials indicating a potential lack of understanding the task. All behavioral data, including the removed participants' data, is available at https://osf.io/rgvf7/. As the purpose of the study was to validate the procedure and the shapes, we included as much data as possible in the analysis. For response time we report data from Correct Trials only and when appropriate, data is collapsed across Trial Type (i.e., Same, Different). We felt that analyzing only Same Trials eliminated too many trials to allow for a true evaluation of our new procedure and stimuli. For example, if we excluded Different Trials, the analysis would come from 16 of the 32 trials and include only 4 of each shape at each angle.

### Data analysis
We used a two-way repeated measure ANOVA with a Greenhouse-Geisser correction with angle of rotation (0°, 50°, 100°, & 150°) and trial type (Same, Different) as factors and found that response time (RT) increased with angle of rotation, $F(1.99,51.79) = 15.49$, $p < .0001$, $\eta_p^2 = .37$ (see Figure 4). Participants responded slower on Different Trials, $F(1,26) = 14.88$, $p = .001$, $\eta_p^2 = .36$, and this effect varied by angle of rotation, $F(2.6,67.68) = 7.38$, $p < .0001$, $\eta_p^2 = .22$ (see Figure 5). Additionally, a linear contrast indicated that RTs increased linearly with angle of rotation $F(1,136) = 13.45$, $p < .001$, $\eta_p^2 = .09$.

A separate two-way repeated measure ANOVA with the same factors showed that Error Rate (ER) also increased with angle of rotation $F(3,102) = 3.69$, $p = .014$, $\eta_p^2 = .10$ (see Figure 3). Participants were more accurate on Same Trials $F(1,34) = 10.34$, $p < .003$, $\eta_p^2 = .23$ and a significant Trial Type by Angle interaction $F(3,102) = 9.64$, $p < .0001$, $\eta_p^2 = .22$ revealed that participants were more accurate on Same Trials compared to Different Trials at the smallest angles of rotation but were less accurate on both trial types at the largest angles (see Figure 4). A linear contrast showed that, like RTs, ERs increased in a linear manner with angle of rotation, $F(1,36) = 6.27$, $p = .017$ $\eta_p^2 = .15$.

The equation of the best-fit regression line for RTs as a function of angle of rotation was y = 1114x + 5514.8, meaning that RTs increased by approximately 22 ms per degree. This means that participants had on average a mental rotation speed of approximately 45° per second. The equation of the best-fit regression line for ERs is y = 0.03x + 0.18, meaning that ERs increased by about 0.03% per degree. This increase in ER is less than other mental rotation studies but the error range for the current study started much higher (even for the lowest angles) and had a smaller increase in error rate than

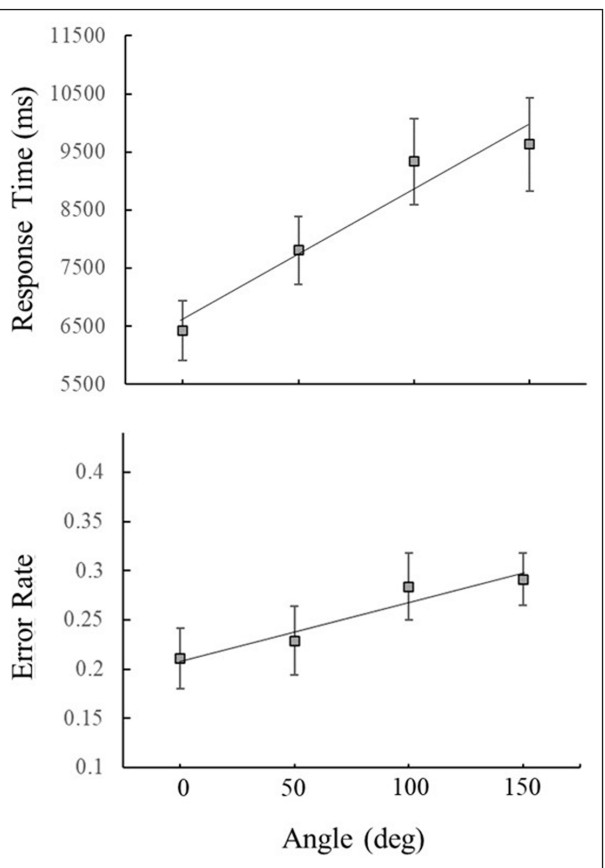

**Figure 4** Mean response times (top) and error rates as a function of angular disparity between sample and test shapes collapsed across Same and Different Trials. Error bars represent the standard error. For response time, only data from Correct Trials was included.

Wolf and Larsen *Journal of Open Psychology Data* DOI: 10.5334/jopd.99     

mental rotation studies, highlighting the difficulty of simultaneous cross-modal mental rotation.

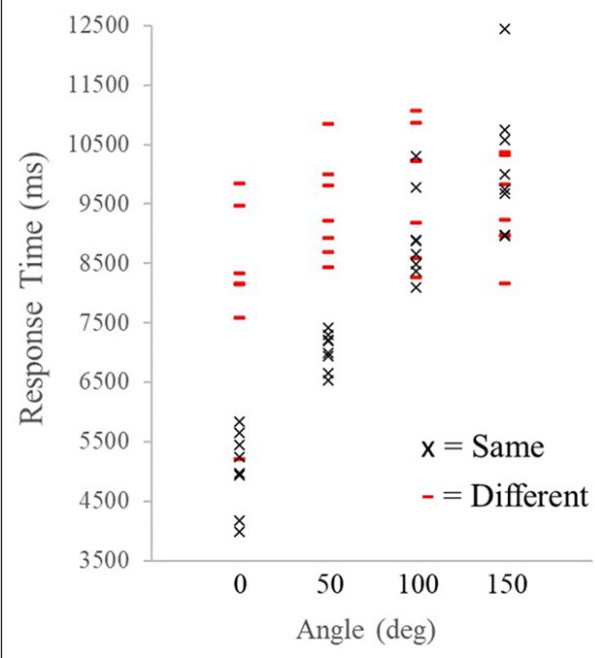

**Figure 5** Mean response times for each of the eight stimuli used in the study as a function of angular disparity and broken down by trial type.

## Discussion

Data collected for the current experiment validated both the new simultaneous cross-modal mental rotation procedure described above and the use of the 3-D printed versions of the shapes inspired by the original Shepard and Metzler (1971) study, designed by Ganis and Kevit (2015) for use as tactile shapes in a mental rotation procedure. As in other studies of mental rotation (e.g., Shepard & Metzler, 1971, 1974; Ganis & Kevit, 2015) response times and error rates increased in the expected linear fashion as angular disparity increased and participants responded slower on Different Trials than on Same Trials. While these results are consistent with previous mental rotation studies, participants' mental rotation was slower (i.e., approximately 42°/ second) than participants mental rotation in the original Shepard and Metzler (1971) study (i.e., approximately 60°/second) or the study conducted by Ganis and Kievit (2015) (i.e., approximately 100°/second), both of which utilized unimodal comparison. However, this slower rotation is not unexpected, as other research relying on cross-modal comparison during metal rotation report performance deficits and slower reactions times as well (Ernst et al., 2007; Miller & Sheinberg, 2022; Norman et al., 2004, Norman et al., 2008; Pamplona et al., 2022). While error rates increased in a mostly linear fashion with angle of rotation, participants in the current cross-modal task were less accurate (i.e., average accuracy at

0° < 80%) than participants in traditional (i.e., non-tactile or cross-modal) tasks (i.e., average accuracy at 0° > 94%) (Ganis & Kievit, 2015).

### 2.7 DATA ANONYMISATION AND ETHICAL ISSUES

All participants completed an informed consent document prior to the start of the study. The data from each participant was randomly assigned a number to anonymise the data and participants were informed that the collected data had the potential to be used in published research. All aspects of the research study were approved by Carroll University's Institutional Review Board. IRB # 19–025. Recorded demographic information (i.e., participant #, age, sex, year in school, & major) cannot be combined to identify individual participants as the participant number is only attached to the data. Also, within each demographic group there are multiple participants in each category (e.g., age, major, sex, & year in school).

### 2.8 EXISTING USE OF DATA

No publication or outputs of this data at the time of this submission.

## (3) DATASET DESCRIPTION AND ACCESS

### 3.1 REPOSITORY LOCATION

https://osf.io/rgvf7/
DOI 10.17605/OSF.IO/RGVF7

### 3.2 SHAPE/FILE NAME

- Within the folder labelled "All Shapes STL files" there are 96 .stl files labeled: shapenumber.stl or shapenumber-R.stl (e.g., 1.stl, 1-R.stl through 48.stl, 48-R.stl)
- Within the folder labelled "Raw and Processed Individual Subject Data" folder there are 74 additional .csv files with each participant's data separated into raw and processed files labeled: P1 processed.csv, P1 raw.csv, P2 processed.csv, P2 raw. csv etc.
- Within the folder labeled "SPSS Syntax Files for Data Analysis" there are syntax and .txt files for the analyses run on the data.

### Additional Files

- Individual Subject Data Description.txt
- Mental Rotation Experimenter Instruction Script.docx
- Mental Rotation Experimenter Instruction Script.txt
- Participant Demographics Combined.csv
- Shapes STL description.txt

### 3.3 DATA TYPE

Data provided is in a raw form with some formulas included for mean and standard deviation of errors committed.

### 3.4 FORMAT NAMES AND VERSIONS

Data files are saved as .csv files and can be opened with many types of software. 3-D printing files are saved as STL files. STL files can be opened with free to download 3-D printing/modeling software, CURA, from UltiMaker (available for Mac, Windows & Linux operating systems @ https://ultimaker.com/software/ultimaker-cura/). The written script of instructions used by the experimenters facilitating the study is saved as a Microsoft Word .docx file and as an .txt file for accessibility purposes.

### 3.5 LANGUAGE

All data and descriptions of data are stored/written in American English.

### 3.6 LICENSE

CC-By Attribution 4.0 International

### 3.7 LIMITS TO SHARING

No limits to sharing.

### 3.8 PUBLICATION DATE

Published on August 24th, 2023.

### 3.9 FAIR DATA/CODEBOOK

The "Individual Subject Data Description" file uploaded to the OSF repository serves as the codebook for data files used in this study and uploaded to the OSF repository.

#### Findable

All data have been assigned unique identifiers and are described in clear and simple American English and are available on the Open Science Framework repository. The selected keywords should help the data reach a wide audience and yet be specific enough for use within the cognitive psychology and mental rotation fields.

#### Accessible

The freely available data files can be viewed with common computer software and are saved as .csv files. The instruction set used by the researchers for implementation of the procedure is also available in a .txt file. STL files can be opened with free to download 3-D printing/modeling software, CURA, from UltiMaker (available for Mac, Windows & Linux operating systems @ https://ultimaker.com/software/ultimaker-cura/).

#### Interoperable

Data are in a format that allows for direct integration into other data sets or are easily modifiable to fit other data set formats. Data are stored and labeled with commonly used column headings.

#### Reusable

Data are stored and labeled according to domain-relevant standards. These data are released in a fully open format with no limit to access.

## (4) REUSE POTENTIAL

The raw data can also be reused in re-running the analyses if future researchers want to reproduce the study or be used in the running of new analyses to test the robustness of the results or answer new research questions. Additionally, due to their highly modifiable nature (e.g., 3-D filament color, size, texture, material, etc.) the stimuli and the STL files from which they were produced can be used not only by researchers interested in cross-modal mental rotation studies but for other types of research (e.g., touch only mental rotation or touch research not limited to mental rotation) that would benefit from the use of the highly modifiable and complex 3-D shapes. The possible sizes, material, and textures of these shapes create the potential for a nearly endless stimulus set for future research to answer questions that have not yet been asked, or those questions currently limited by practice effects with the same or limited mental rotation shapes currently available.

## ACKNOWLEDGEMENTS

We thank Isabelle Banke, Amanda Borchardt, Brianne Ford, Morgan Guidry, Emilie James, Amelia Thiry, Zachary Weiss, and Leah Witthuhn for their contributions to the project, including literature searching, experiment design, and data collection. We also thank Dr. Meghan Dowell who was an important source of 3-D print knowledge, assistance and suggestions during the shape design and printing process.

## COMPETING INTERESTS

The authors have no competing interests to declare.

## AUTHOR CONTRIBUTIONS

Joshua Wolf oversaw development of project, collection of the data and performed the analysis of the data. Joshua Wolf wrote the manuscript with the help of Melissa Larsen, who provided critical feedback and support during the analysis of the data, and all stages of manuscript preparation process.

## AUTHOR AFFILIATIONS

**Joshua E. Wolf** (iD) orcid.org/0009-0009-8548-1889
Carroll University, 100 N East Avenue, Waukesha, WI 53186, USA
**Melissa Larsen** (iD) orcid.org/0009-0007-4550-2740
Carroll University, 100 N East Avenue, Waukesha, WI 53186, USA

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

## PEER REVIEW COMMENTS

*Journal of Open Psychology Data* has blind peer review, which is unblinded upon article acceptance. The editorial history of this article can be downloaded here:

- **PR File 1.** Peer Review History. DOI: https://doi.org/10.5334/jopd.99.pr1

**TO CITE THIS ARTICLE:**

**Submitted:** 25 August 2023    **Accepted:** 03 April 2024    **Published:** 24 April 2024

