## [Peer Review History. · Journal of Open Psychology Data]

Review of A new procedure and stimulus set for examining cross-modality mental rotation.

The manuscript is a detailed description of a cross-modal mental rotation stimulus set and a report of the results from a study using described stimuli. In the study, participants were comparing two 3D-printed figures, one presented visually and one, which they could only explore with touch to determine if the shapes were the same or different. The study results revealed a linear relationship between the angle of rotation and response times as well as error rates, similar to other mental rotation studies. All data and stimuli from the study are freely available for reuse. This is an important contribution to the mental rotation research as well as to cognitive psychology more generally.

Here are my suggestions for improving the manuscript and the available set of data and study materials:

- Paragraph “2.4 Sampling, sample and data collection” could be changed to “2.4 Participants” and “2.5 Materials/ Survey instruments” to simply “2.5 Materials”
- Figure 2 could be more helpful for the reuse of the study materials if the exact labels and dimensions were provided within the figure (either in Figure 2 in the manuscript or included as an additional supplementary figure in case the picture will get too crowded with all labels and dimensions). For example, it would be useful to know the exact chamber dimensions, where the “sample shape line” (mentioned in the manuscript) is and which element is the “divider”.
- I can't locate "Combined_Subject_Data" in the OSF repository, a file that was supposed to contain demographic information about study participants.
- In “Individual Subject Data Description” file there is brief information about each variable, but it could be helpful to expand the descriptions. One example could be expanding on what is "Stimulus order running sheet designation". Providing more details about each variable would make sure they will be well understood by those who would like to reuse the data.
- In the Discussion, the authors speculate that “The slower rotation by participants in the current study may be due to the difficult nature of the simultaneous comparison across modalities and the lack of cube segmentation or other features that allow for quick and accurate 3-D representation. It seems much more likely that this simultaneous cross-modal comparison is much more difficult than standard mental rotation procedures with traditionally viewed stimuli.” It could be helpful to try to find some support for this speculation in the literature on cross-modal integration and refer to appropriate sources. I am not an expert in this field, but I would assume there is reported evidence for differences in behavioral responses whenever cross-modal integration needs to happen compared to when there is no need for such integration.
- For “2.7 Data anonymisation and ethical issues” - If participants were informed that anonymous data from this study would be made openly available, that would be good to report here. I cannot locate the demographics file (see earlier comment) but it would be also important to double check that no demographics information combined can identify individual subjects from this study.
- In “Additional files”, the demographics file is not listed (see earlier comment).
- Regarding formats names and versions, Excel is a proprietary software (i.e. not everyone will have access to it for free). Is it possible to make .xlsx files available in .csv format (which is an open, non-proprietary format that can be opened in many

different types of software)? Similarly, could the script with study instructions be saved in .txt file in addition to .docx file for accessibility? (Bolded text can be changed to all caps in order to differentiate between the study script parts). For 3D-software, could the exact link/resource be provided if it's freely accessible and downloadable software?

- Regarding files with participant's responses, as much as it is helpful to see the calculations for each participant in each data file, it would be difficult to import and merge files in this format in many types of statistical software. (e.g. for reproducing the analyses or running new analyses on available data). It could be easier to have only raw data in participant files and processed data (i.e., calculations of averages) in separate files.
- If data analyses for the reported study were conducted in SPSS, could you provide and made openly available the syntax file from SPSS (i.e., the step-by-step syntax of how analyses were performed). This is both for transparency of the analyses but also for reproducibility in case others would like to re-run the analysis or build on the performed analyses.
- The word "script" is often used for data analyses these days (e.g. programming script), so it could be helpful to disambiguate experimenter's script by saying: "written script of the study instructions used by.." (in "3.4 Format names and versions")
- In "3.9 FAIR data/codebook" it says that there is no codebook; however I think that the "Individual Subject Data Description" file serves as codebook and should be referred to there.
- In "Findable" section, change "Open Science Forum" to "Open Science Framework" (if you meant a different platform that OSF, please provide more information).
- In "Accessible" section, when you say "common computer software" I assume this applies only to data files (i.e., participant data from the study)? If this applies also to stimuli - the software for opening and editing .stl files is maybe not as common? In this case, you could add: "if files are not accessible through common computer software, we provide information on how to access the files, e.g. through acquiring freely available software." (or similar)
- In "(4) Reuse potential" when you mention reusability of raw data – I think there is more potential in the data itself than just reuse for the purpose of meta-analyses. It may be worth mentioning that raw data could be also reused in re-running the analyses (e.g. when reproducing the study) or running new analyses on the data (e.g. when testing how robust the results are or when answering new research questions with the use of the provided data).
- In the same paragraph, I am not sure what is meant by this part of the sentence: "but also for research with tangible shapes or touch only mental rotation using complex stimuli" (it might be a typo/mistake?).

Finally, I would like to thank the authors for their important contribution to the field and for making all stimuli and data available. I am looking forward to seeing this work published.

Agata Bochynska, PhD
University of Oslo
Oslo, Norway